# Gender Differences in Head and Neck Posture Among Smartphone Users While Walking: Insights from Field Observations in Taipei

**DOI:** 10.3390/jfmk9040245

**Published:** 2024-11-21

**Authors:** Yi-Lang Chen, Ting-Hsuan Wang, Wei-An Chang, Hong-Tam Nguyen

**Affiliations:** Department of Industrial Engineering and Management, Ming Chi University of Technology, New Taipei 243303, Taiwan; u08217102@mail2.mcut.edu.tw (T.-H.W.); u08217129@mail2.mcut.edu.tw (W.-A.C.); m05218010@mail2.mcut.edu.tw (H.-T.N.)

**Keywords:** filed observation, smartphone user, walking, head and neck flexions, gaze angle, viewing distance

## Abstract

**Background/Objectives**: Despite the increasing prevalence of smartphone use while walking, few studies have comprehensively investigated head and neck posture in real-world settings. This study employed a single-blind observation of smartphone users walking in public areas of Taipei, Taiwan, to examine head and neck movements, with a particular focus on the effects of gender and hand-operation type. **Methods**: We conducted observations of 120 smartphone users (60 males and 60 females), recording neck flexion (NF), head flexion (HF), gaze angle (GA), and viewing distance (VD) in the sagittal plane during walking. The analysis included four combinations of gender and smartphone hand operation (one-handed and two-handed use). **Results**: Significant gender differences were found in NF (*p* < 0.001), GA (*p* < 0.01), and VD (*p* < 0.01), with males exhibiting greater NF, GA, and VD than females. HF was similar between genders, but males’ larger NF suggested a more pronounced forward head posture, potentially increasing neck and shoulder strain. Hand operations also significantly affected VD (*p* < 0.001). Most users displayed a GA exceeding 60°, possibly an involuntary adjustment for better visibility of the walking path, potentially increasing eyestrain. **Conclusions**: Previous studies have primarily simulated smartphone use while walking in controlled environments, such as on treadmills or in laboratories. In contrast, our single-blind field study highlights the real-world risks associated with smartphone use during walking, including neck and shoulder strain and eyestrain, with notable differences observed between genders.

## 1. Introduction

Smartphones have become an indispensable tool in our daily lives. In Hong Kong, individuals spend an average of 2.4 h per day using mobile devices, with 20% of users dedicating more than 4 h daily to their smartphones [1]. In Bangladesh, a significant 36.8% of smartphone users allocate 5–10 h to their smartphones, and 9.8% even exceed 10 h of usage [2]. Meanwhile, in Taiwan, there were a staggering 30 million cellular mobile connections at the start of 2023, and an impressive 95.8% of the population accesses the internet through their smartphones for over 8 h each day, surpassing the global average of 7 h [3]. The prevalence of smartphone usage across different countries highlights its widespread influence on modern living.

In recent years, the prevalence of smartphone usage while walking has significantly increased. Many individuals find it exceptionally convenient to use their smartphones while on the move, allowing them to stay connected, access information, or engage in entertainment activities seamlessly. A recent survey conducted by the American Academy of Orthopaedic Surgeons (AAOS) revealed that approximately one-third of smartphone users frequently engage in non-speech activities, such as browsing, texting, or gaming while walking [1]. Similarly, in Taiwan, nearly 40% of smartphone users have developed the habit of using their phones while walking [4]. This trend underscores the growing prevalence of smartphone use in dynamic environments and its impact on daily behaviors.

Over the past years, research on this topic has predominantly occurred in controlled laboratory settings, with a primary focus on static standing and seated postures [5,6,7,8,9,10,11]. The ergonomic assessment of smartphone usage has primarily concentrated on examining the extent of forward head flexion (HF) and neck flexion (NF) during various smartphone activities, aiming to propose strategies to reduce the strain on the neck and shoulders [7,9]. Discomfort in the neck and shoulder area resulting from smartphone usage has been documented in numerous countries and regions [12,13,14]. These findings convincingly establish a clear cause-and-effect relationship between smartphone use and discomfort in the neck and shoulder region.

Within the restricted scope of observational studies in this field, Chen et al. [15] explored HF and NF angles in sitting and standing positions among 600 smartphone users, yet they did not investigate smartphone use while walking. Conversely, Kim et al. [16] discovered that, in comparison to walking without a smartphone, participants showed a slower cadence and velocity, changes in stride length, gait cycle, and increased time in ground contact while using a smartphone during walking. Similar changes were also observed when engaging in texting without an actual task. Walking and texting seemed to result in reduced local dynamic stability and increased regularity. These observed changes may indicate compromised balance while texting during walking, regardless of walking speed [17], and could potentially elevate the risk of slips, trips, and falls [18,19].

The majority of studies investigating changes in head and neck movement during smartphone use while walking have predominantly relied on laboratory experiments [17,20,21,22]. Yoon et al. discovered that the activation of neck–shoulder muscles during smartphone use while walking exceeded that observed during sitting and standing [22]. Whether participants were browsing or texting, the NF of smartphone users was significantly lower while walking than when standing. Additionally, participants tended to have a more downward gaze when using smartphones while walking compared to when sitting [1]. These findings collectively underscore the importance of considering the dynamic aspects of smartphone use and its influence on postural characteristics during walking, with implications for ergonomics and user safety. Research on one-handed browsing and two-handed texting tasks has traditionally concentrated on the static posture of the neck area, assuming that two-handed operation results in a relatively balanced and low load, providing high stability for holding the mobile phone [23]. In contrast, one-handed manipulation typically involves a smaller cervical flexion angle [6,24] but increased muscle activity in the upper arm [25]. A study by Han and Shin [20] also revealed that when individuals texted with two hands while walking, the HF angle of smartphone users was significantly larger than when browsing with one hand. Variations in the results among these studies may be attributed to differences in experimental settings.

The increasing prevalence of smartphone usage has raised concerns regarding ophthalmic issues associated with prolonged use [26]. An essential factor in smartphone operation is the viewing distance (VD). Bababekova et al. [27] found that the average VDs for reading messages and browsing the internet on smartphones were significantly shorter than typical VDs for other electronic devices [28]. Viewing a screen or target at such close ranges intensifies visual demands, including accommodation and convergence, potentially increasing intraocular pressure [29] and exacerbating symptoms of asthenopia. To ensure a comfortable and healthy range for head and neck posture during smartphone use, it is generally advisable for the associated gaze angle (GA), representing the user’s line of sight to the horizontal plane, to fall between 40° and 60° [30]. However, assessments of GA and VD during smartphone operation have primarily been limited to static standing or sitting positions. Additional research is essential to scrutinize these dimensions of smartphone use in dynamic situations like walking, aiming to gain a comprehensive understanding of the potential impacts on musculoskeletal well-being and visual health.

When investigating smartphone usage behaviors during walking, previous research has mainly involved instructing users to walk a specific distance at their usual or preferred pace [19,20,21]. Nonetheless, some researchers have chosen laboratory experiments that incorporate treadmill-based testing [1,17,22,31]. However, no prior research has delved into the examination of head/neck posture in real-world settings. Conducting a single-blind field observation offers a more authentic portrayal of users’ natural postures while using smartphones. In this field observation, our study attempted to collect NF, HF, GA, and VD data for a total of 120 participants, including 60 male and 60 female users, while they engaged in either one-handed or two-handed tasks while walking. The primary objective of this study was to examine the impacts of gender and task variables, such as hand operation type, on head and neck posture movements during smartphone use while walking. We hypothesized that these factors would significantly influence neck and head movements. By employing a single-blind field study design, we aimed to provide a more realistic basis for understanding these effects, offering insights that build upon prior research conducted in controlled laboratory settings.

## 2. Materials and Methods

This study primarily centered on exploring the differences in head and neck postures within four test combinations involving two genders and two hand operations during smartphone use while walking in the field. The experiment was conducted from March to April 2024, during daylight hours with random scheduling, excluding periods of poor visibility at night. The research followed the ethical principles outlined in the 2013 World Medical Association Declaration of Helsinki and received approval from the Research Ethics Committee of National Taiwan University in Taiwan (code: NTU-REC-202312-EM-051). Informed consent from all participants was obtained and validated for the publication of identifying information/images in an online open-access platform.

### 2.1. Participants

A total of 120 smartphone users, evenly distributed with 60 men and 60 women, were selected at random and observed in various public areas throughout Taipei, Taiwan. These locations included university campuses, subway stations, sidewalks, and parks. A priori power analysis using G*power (Version 3.1.9.7) for the fixed effects of ANOVA was calculated beforehand and resulted in 111 subjects. A power of (1 − β) = 0.80, an α-error probability of 0.05, and a large effect size of 0.4 were assumed. Our actual sample of 120 participants thus exceeded the minimum requirement for statistical power. The study recorded the mean (standard deviation, SD) age, height, and body mass of each subgroup, categorized by one-handed or two-handed smartphone operation, as shown in Table 1. No significant differences were found in these individual characteristics between the paired subgroups. All users provided written consent and their basic information, including smartphone size, only if they willingly agreed to participate. In cases where users declined to take part, their photos were promptly deleted.

### 2.2. Posture Measurement

This study assessed NF, HF, GA, and VD while participants were walking and using smartphones. While direct assessments of neck–spine positions with the precision of wireless IMUs (Inertial Motion Units) used in previous studies [20,32] are not currently available, issues regarding measurement validity and limitations still require clarification [33]. Moreover, the use of this specific method was not feasible in our single-blind test. To obtain these measurements, we captured symmetrical sagittal postures and used CorelDRAW (Corel Co., Ottawa, ON, Canada, Graphics Suite, 2023 version 24.5) for digital markings. We identified key landmarks, including the seventh cervical (C7) and thoracic (T7) spinous processes, canthus, and tragus, and marked half of the smartphone’s length in the photos, as shown in Figure 1. Subsequently, we calculated the upper thoracic angle (UTA), NF, HF, GA, and VD. Following the approach proposed by Guan et al. [30], we used UTA, defined as the angle between the line from C7 to T7 and a vertical line, to measure NF in this study. This method was found to be practical since the upper thorax was considered a rigid body. The measurement of UTA and the corresponding NF followed a method similar to that employed by Chen et al. [15] and Yoon et al. [22].

To validate the field measurements for both their accuracy and consistency, as previously conducted by Chen et al. [15], we employed a camera (resolution: 1:30,000 at 60 Hz) and also utilized a motion capture system (Qualisys MacReflex, Göteborg, Sweden) to corroborate the measurements derived from the CorelDRAW software utilized in the study. An additional group of 10 participants (5 males and 5 females), attired in summer clothing as observed in the field study, were instructed to engage in natural walking while using their smartphones, with or without body joint markers attached. The postures of these participants were captured using a motion analysis system and subsequently photographed. Joint positions were identified, and angles and VD were estimated using the CorelDRAW software by three experimenters. The data derived from CorelDRAW were then cross-referenced against the data obtained from the MacReflex system. Three experimenters photographed user postures and subsequently recorded the measurement data. To ensure angles and distances were consistent among photos taken by different experimenters, the intraobserver and interobserver reliabilities were determined. The mean absolute difference was also used to examine differences in the estimated angles and distance for each observer or among the observers. Forty samples were examined (10 samples for each gender and posture combination), and the three experimenters conducted repeated measurements (with two measurement intervals of >6 h) for reliability analyses. As a result, the recorded measurement errors consistently fell within a margin of 2.2° for angles and 1.5 mm for distances, confirming the validity of the measurements. The intraclass correlation coefficients for the four investigated measures were 0.921–0.985, demonstrating high internal consistency. The interclass correlation coefficient between any two observers was 0.908–0.954, indicating good reliabilities among the three experimenters.

### 2.3. Design and Procedure

The field observations involved the collection of NF, HF, GA, and VD data for four distinct smartphone use scenarios, categorized by gender and the type of operation (as depicted in Figure 2). In total, 120 participants were randomly selected and observed during the study. Once 30 qualified samples were successfully collected for any particular combination, sampling for that specific scenario was considered complete. For the field photography, we used cameras (Sony, ZV-E10 + 16-50MM KIT, Tokyo, Japan) positioned at a predetermined location along a walkway, situated 6 m from the camera. The camera height was adjusted to approximately align with the shoulder height of the smartphone users. Images were taken for analysis when the participants walked perpendicular to the camera’s line of sight in the video. However, prospective participants were required to maintain an approximately normal walking speed, as assessed by the experimenter’s visual judgment. Following the photography session, users were approached and asked for their willingness to participate in the study. Those who agreed signed a consent form and provided basic personal information, including age, height, and body dimensions. Additionally, the researchers recorded the phone length for normalizing VD. Photos of users who declined to participate were promptly deleted. Fortunately, a significant number of users (86%) willingly participated, possibly attributable to the prevailing habit of wearing masks in public areas during the post-COVID-19 pandemic stage, potentially mitigating privacy concerns.

In the analysis, outlier photographs were excluded from the data, including images of users carrying large backpacks, heavy objects exceeding 5% of their body weight [34], displaying unnatural walking postures, or having difficulties in identifying body landmarks. The unconventional protocol of capturing photographs before obtaining consent was employed to avoid observation bias and to capture users’ genuine natural postures during smartphone use. Subsequently, CorelDRAW software was utilized to identify and mark anatomical landmarks on the photos (as shown in Figure 2), facilitating angle and distance measurements. In the analysis, we used the average static posture of a sample of 30 individuals to represent the dynamic posture for each test combination.

### 2.4. Statistical Analysis

The data collected from the test were analyzed using SPSS 23.0 statistical software (IBM Corp., Armork, NY, USA), with a significance level set at 0.05. The study aimed to investigate the effects of participant gender (men and women) and hand operations (one- and two-hand) on the measured responses (NF, HF, GA, and VD) through a two-way analysis of variance (ANOVA). In the analyses, gender and hand operation were treated as between-subject factors due to the nature of the field observation study. Additionally, to determine the practical importance of any significant independent variable, the power value was calculated based on Cohen’s guidelines [35]. Since the comparative groups had identical sample sizes, the effect size was calculated by subtracting the group means and dividing the result by the pooled standard deviation. This effect size represents the magnitude of the difference between the groups in terms of their shared standard deviation. An effect size of ≥0.2 indicates a small effect, ≥0.5 indicates a medium effect, and ≥0.8 a large effect. Beforehand, the Kolmogorov–Smirnov test was utilized to assess the conformity of numerical variables with the normal distribution, while Levene’s test was employed to assess the homogeneity of variances.

## 3. Results

Table 2 displays the results of the two-way ANOVA. With the exception of HF, the gender variable had a significant impact on NF (*p* < 0.001), GA (*p* < 0.001), and VD (*p* < 0.01). Hand operation only had a significant effect on VD (*p* < 0.001). The absence of a significant two-way interaction effect (all *p* > 0.05) suggests that the main effects of the gender and hand operation variables have been confirmed.

Figure 3 and Figure 4 illustrate the differences in the main effects for the paired levels of gender and hand operation, respectively. The figures reveal that male participants exhibited larger values for NF, GA, and VD than female participants, and additionally, we observed larger VD during two-handed texting compared to one-handed browsing. When averaged across the hand operation variable, males’ NF, GA, and VD were 29.3°, 66.1°, and 38 cm, respectively, while the corresponding values for females were 23.9°, 61.0°, and 34.9 cm. The difference in VD between the two hand operation methods across genders was 5.4 cm.

## 4. Discussion

Previous research has predominantly focused on static smartphone postures, such as standing or sitting, with increasing attention to dynamic walking postures studied in controlled laboratory settings. In contrast, this single-blind observational study explored head and neck postures during smartphone use while walking in real-world conditions. Consistent with expectations, the findings demonstrated that gender significantly influenced NF, GA, and VD, with males showing greater NF, GA, and VD compared to females. The larger NF observed in males indicated a more pronounced forward head posture, potentially increasing neck and shoulder strain. Additionally, hand operation type significantly affected VD. Notably, most users demonstrated a GA exceeding 60°, likely an involuntary adjustment to improve the visibility of the walking path, which could contribute to increased eyestrain.

The dependent variables of NF, GA, and VD exhibited notable gender-related differences (Table 2). Females demonstrated lower NF, GA, and a shorter VD (Figure 3), consistent with prior research on postures during standing and sitting [36,37]. Notably, while females exhibited smaller NF than males (23.9° vs. 29.3°) during smartphone use while walking, HF values were nearly identical between genders (93.9° vs. 94.5°). In previous research that primarily focused on walking [20,22], there was no study comparing gender differences in head/neck postures during smartphone use. However, Guan et al. [37] found that when standing and using phones, both NF and HF of female participants were significantly smaller than those of males. Interestingly, their recorded GAs were identical between genders (59.4° vs. 60.1°), whereas our measurements differed (61.0° vs. 66.1°). These variations can be attributed in part to the inherent distinctions between standing and walking postures. However, some of the differences might be due to variations between field and laboratory test settings. In our study, participants walked in public areas with a natural posture, unaware that they were being photographed. During this period, participants likely had to stay attentive to their surroundings, including avoiding potential collisions with pedestrians. This heightened awareness could contribute to their smaller NF and, consequently, a larger GA. In contrast, in laboratory settings, participants might not have faced the same environmental distractions.

In contrast to the identical HF, male participants in this study displayed a significantly larger NF than their female counterparts. This suggests that males may adopt a more pronounced forward head posture (FHP), commonly known as “text neck” or “turtle neck” [30,38,39], when walking and using smartphones in public areas. To clarify, NF typically refers to the forward flexion of both the head and cervical spine as a single unit, while HF refers to head bending with the upper cervical spine serving as the axis of rotation. Indeed, the mechanism of forward head and neck flexion varies fundamentally [30,38]. During walking in public areas, both male and female participants maintained an HF of approximately 94°. This angle might be the posture chosen by the participants when browsing or texting on smartphones while simultaneously being mindful of their surroundings. However, male participants exhibited an NF 5.4° larger than that of female participants, which could potentially lead to increased strain on the neck.

When comparing the findings of this study with a similar field study conducted by Chen et al. [15], notable differences in HF and NF values emerge. Chen et al.’s research, which examined 300 men and 300 women using smartphones in a stationary standing posture, reported HF angles of 101.7° for males and 98.2° for females. These values are higher than the HF angles observed in our study, where both male and female participants displayed HF values around 94°. The reduced HF in our study is likely due to the dynamic nature of walking, which requires users to adopt a more upright posture to maintain balance and stay aware of their surroundings. Regarding NF, our results also showed smaller angles compared to Chen et al.’s findings. Specifically, Chen et al. reported NF values of 38.6° for males and 36.8° for females, whereas our study found average NF values of 29.3° and 23.9° (calibrated by UTA) for males and females, respectively. These differences align with prior research, such as Lee and Jeon [21], who observed NF angles of approximately 30.1° across genders during walking. The slightly higher NF values reported in their study may be attributed to the controlled laboratory setting, where participants likely faced fewer external distractions than in real-world environments. The smaller NF observed in our study is consistent with findings by Yoon et al. [22], who demonstrated that walking while using a smartphone generally results in reduced NF but increased muscle activity in the associated muscle groups. This reduction in NF during walking is likely due to the need for users to maintain situational awareness and adjust their posture to avoid potential hazards, unlike the more relaxed posture adopted during stationary standing. These findings emphasize the importance of context when analyzing head and neck postures during smartphone use. Dynamic environments, such as walking, impose unique constraints that alter postural characteristics, reducing both HF and NF compared to static conditions.

Our analysis revealed that females exhibited a lower GA and a shorter VD (Figure 3), consistent with prior field observation research [15] on postures during standing and sitting. The average VD for females in this study was 34.9 cm, significantly shorter than that of males (38.0 cm), in line with earlier findings [15,27,40]. Typically, females have shorter forearms, leading to a correspondingly shorter VD in comparison to males [40]. It is worth noting that a shorter VD can potentially exacerbate eyestrain symptoms [27,41,42], although whether this effect varies based on gender differences warrants further investigation. VD, from the perspective of human posture characteristics, can be regarded as an outcome of the coordination between NF and the position of holding phones. Our observations indicate that female participants exhibited a straighter neck posture (i.e., a smaller NF) but a shorter VD compared to male participants in the test. While differences in VD may be partly attributed to anthropometric variations between sexes, NF differences are unrelated to sex [43]. Given women’s common tendency to adopt more intentionally upright postures [44] and the need to raise the phone to view the screen, this may lead to increased engagement of the neck and shoulder muscles in female smartphone users. Supporting this idea, Chen et al. [45] found that cervical erector spinae and upper trapezius activity were significantly higher in women. This increased muscle activity suggests that women may experience greater strain on their neck and shoulder muscles when using smartphones.

To maintain a comfortable head and neck position, Guan et al. [37] recommended maintaining a GA between 40° and 60° when using smartphones. Figure 3 indicates that the average GA for males and females was 66.1° and 61.0°, respectively, with over half of the users exceeding the recommended safe range. The reason for the larger GA may be that users have to limit excessive neck and head flexion while walking, thereby increasing the GA. This was more apparent for males because their more significant FHP than females. Importantly, participants did tend to gaze downward more while using smartphones during walking compared to sitting and the change in gaze direction is probably an involuntary reflex to ensure safety by increasing the visual field of the walking path [1]. However, this downward gaze may induce sustained contraction of the extrinsic eyeball muscles, especially the rectus inferior, possibly leading to muscle fatigue [46]. Further investigation is warranted to determine whether an excessively large GA could potentially lead to increased extraocular muscle forces generated during convergence [47] or affect the magnitude of accommodation [48], potentially resulting in increased eyestrain.

In the case of participants using both hands for texting, our study only found a significantly larger VD among the four measures investigated (Figure 4). These results appear somewhat inconsistent with previous studies conducted by Vahedi et al. [23], who noted increased NF during two-handed texting in a standing or sitting position, and Yoon et al. [22], who observed higher HF during two-handed texting while walking compared to one-handed browsing. The differing trends in our study might be attributed to the fact that it was conducted in a field observation setting. While using smartphones, the primary behavior of users in our study was still to pay attention to their surroundings to avoid potential hazards, which prevented them from using their phones with their heads down. When head and neck posture were a safety concern during walking, participants’ VD may depend on their hand position. This effect may be more evident for the two-handed texting task. When using smartphones, participants may adjust their VD within a reasonable range to strike a balance between preventing eyestrain and minimizing upper-arm muscle loads. This aspect requires further clarification. Notably, neither Vahedi et al. [23] nor Yoon et al. [22] investigated the impact of hand operation on GA and VD. Guan et al. [37] suggested that during video viewing, participants did not need both hands, leading them to position their smartphones closer to their eyes. Our findings align with Guan et al.‘s observations, but it is also plausible that our participants made adjustments to their smartphone positions while walking to mitigate the increased HF.

This study has several limitations that should be considered when interpreting the findings. First, the results may not be generalizable to broader populations, as the observations were conducted exclusively in Taipei, Taiwan, and may not reflect smartphone usage behaviors in other regions or cultural contexts. Second, the study utilized a two-dimensional sagittal plane evaluation to assess postures, which does not fully capture the complexity of three-dimensional human movements. Future studies should incorporate three-dimensional motion analysis to provide a more comprehensive understanding of head and neck posture during smartphone use. Although walking is inherently dynamic, this study used the average static posture of a sample of 30 individuals to approximate the dynamic posture for each test combination. Third, the study did not account for additional variables or utilize instruments that could offer deeper insights into participants’ physical and environmental conditions. These include factors such as musculoskeletal health, physical fitness levels, carriage, smartphone model and usage type, walking surface characteristics, crowd density, and lighting conditions. Including these measures in future research could enhance the understanding of the relationship between posture and broader health outcomes. Fourth, the potential influence of participant height as a confounding variable was not explicitly addressed. Variations in height could affect viewing distance and posture, and future studies should consider normalizing such factors to better isolate the effects of gender and smartphone usage behaviors. Finally, as this study focused on short-term posture observations during walking, it did not assess the long-term effects of smartphone use on musculoskeletal health or behavior. Future longitudinal studies could provide valuable insights into the chronic impacts of these behaviors. Addressing these limitations will help refine methodologies and expand the relevance of findings for ergonomic and health recommendations.

## 5. Conclusions

This study investigated head and neck movements among male and female smartphone users during walking in real-world environments, emphasizing the effects of gender and hand operation on NF, HF, GA, and VD. The findings highlighted significant gender-related differences in NF, GA, and VD, with males exhibiting larger values compared to females. For hand operation, only VD was significantly affected, with larger values observed during two-handed operation compared to one-handed operation. No interaction effects between gender and hand operation were found, indicating independent influences on posture. These results provide valuable insights into the postural behaviors of smartphone users in dynamic, real-world settings, contributing to a better understanding of ergonomic risks.

The findings of this study have practical implications for the development of ergonomic guidelines and interventions to reduce neck and shoulder strain associated with smartphone use. Suggested strategies include enhancing posture awareness, designing ergonomic accessories, and implementing educational programs to alleviate musculoskeletal discomfort and eye strain. These efforts can contribute to improving well-being in an increasingly digital society. However, future studies should address the limitations of this research by incorporating three-dimensional motion analysis, investigating additional factors such as physical condition, usage patterns, walking pace, distractions, and environmental conditions, and conducting longitudinal studies to examine the long-term effects on musculoskeletal health. Broader investigations across diverse populations and cultural contexts are also essential to enhance the generalizability of these findings.

## Figures and Tables

**Figure 1 jfmk-09-00245-f001:**
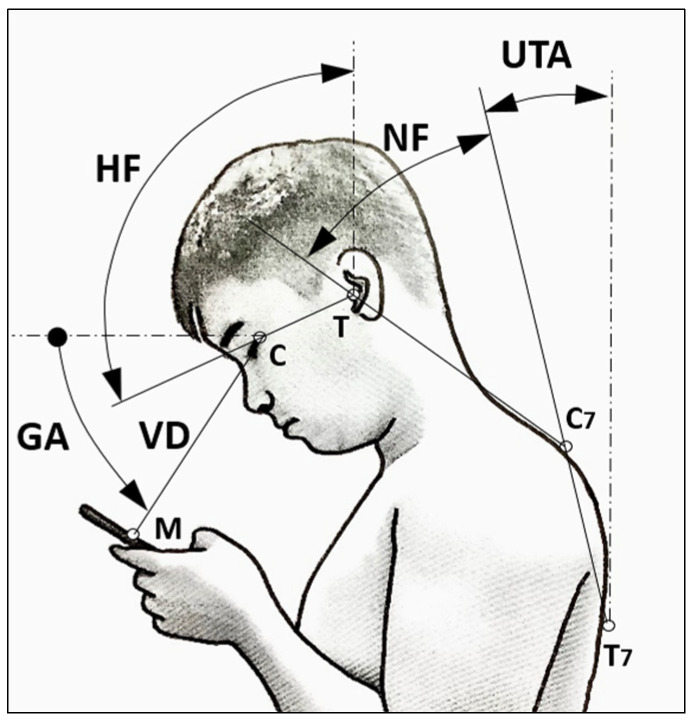
Schematic of the markers and definitions of angles and distances on the human body (notes: HF, head flexion; NF, neck flexion; GA, gaze angle; VD, viewing distance; UTA, upper thoracic angle; T, ragus; C, canthus; M, midpoint of phone length; C7, seventh cervical spinous process; T7, seventh thoracic spinous process).

**Figure 2 jfmk-09-00245-f002:**
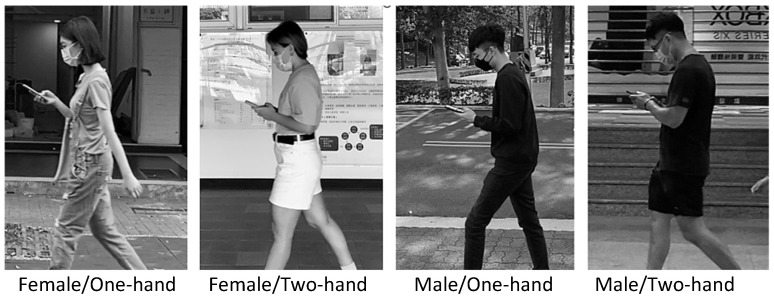
Examples depicting the combinations of gender and hand postures observed during the test.

**Figure 3 jfmk-09-00245-f003:**
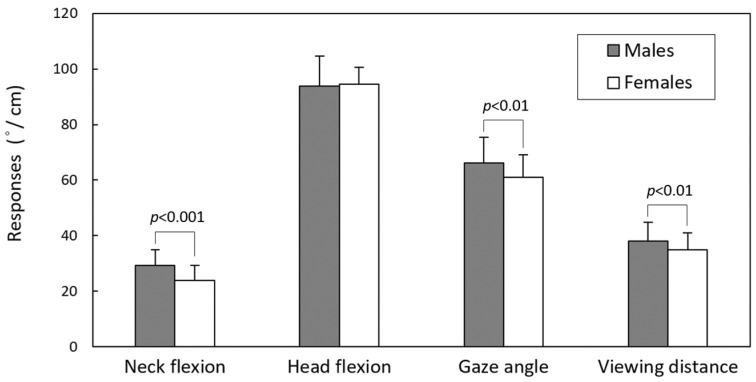
Comparisons of the main effects of all measures between two genders.

**Figure 4 jfmk-09-00245-f004:**
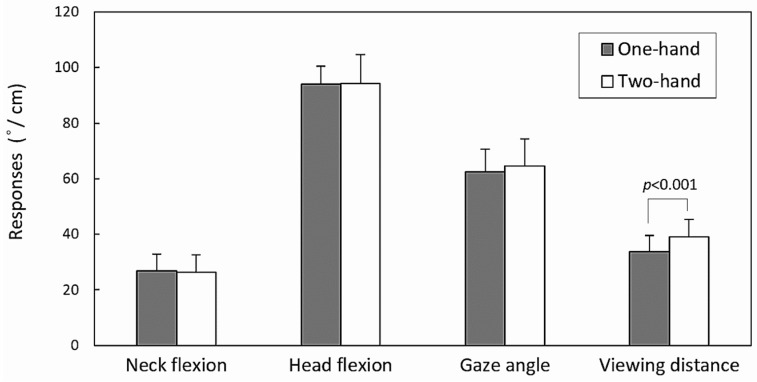
Comparisons of the main effects of all measures between two hand operations.

**Table 1 jfmk-09-00245-t001:** Basic information of male and female participants, categorized by hand operation type.

		One-Hand Operation		Two-Hand Operation	
Gender	Items	Mean	Standard Deviation	Mean	Standard Deviation
Men	Age (years)	26.0	7.0	26.8	7.2
	Height (cm)	172.0	7.3	174.5	6.9
	Body mass (kg)	66.5	5.1	69.6	5.8
Women	Age (years)	26.1	7.7	25.7	8.3
	Height (cm)	158.2	6.2	159.4	6.8
	Body mass (kg)	56.1	5.4	55.6	5.2

Notes: The sample size for each subgroup, categorized by gender and hand operation, was 30.

**Table 2 jfmk-09-00245-t002:** Results of the two-way analysis of variance for four measures.

Variables	Measures	SS	DF	MS	F	*p*-Value	Power
Gender	Neck flexion	861	1	861	27.55	<0.001	0.999
	Head flexion	11	1	11	0.15	0.704	0.066
	Gaze angle	789	1	789	10.42	<0.01	0.893
	Viewing distance	274	1	274	7.77	<0.01	0.790
Hand operation	Neck flexion	12	1	12	0.39	0.534	0.095
	Head flexion	<1	1	<1	<0.01	0.962	0.050
	Gaze angle	115	1	115	1.52	0.221	0.231
	Viewing distance	886	1	886	25.09	<0.001	0.999
Gender × Hand operation	Neck flexion	4	1	4	0.13	0.719	0.065
	Head flexion	8	1	8	0.10	0.751	0.062
	Gaze angle	131	1	131	1.74	0.190	0.257
	Viewing distance	10	1	10	0.30	0.588	0.084

## Data Availability

The data are available upon reasonable request to the corresponding author.

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
