# Peer review of "Gender Differences in Head and Neck Posture Among Smartphone Users While Walking: Insights from Field Observations in Taipei"

_jfmk, 2024, doi:10.3390/jfmk9040245_

Round 1
Reviewer 1 Report
Comments and Suggestions for Authors
First of all, I would like to thank you for being invited to read the document.
The authors have done an excellent job. Each of the comments shared are intended to improve the study.
The comments can be found in the PDF document.
Also, some of the comments on some of the points that need to be reworded in the paper are shared below:
ABSTRACT (Line 9-12)
Revise the structure of the journal to be able to articulate these two sections of the abstract in background/objectives.
INTRODUCTION (Line 104-110)
Revise because it generates confusion, it seems that there are three objectives. The first one is to examine NF, HF, GA and VD. Then to present the hypothesis and finally, the third objective which mentions proposing an alternative research method.
However, there is never any mention of comparisons by gender or by task.
It could be clarified that the study had two objectives, so it would be necessary to modify it in the abstract and in the beginning of the discussion.
PARTICIPANTS (Line 121-123)
There is a power test for the sample size used to define 120 people (60 women and 60 men). If so, it would be important to detail it.
POSTURE MEASUREMENT (Line 166-169)
Developed some statistical test to determine the margin of error.
DESIGN AND PROCEDURE (Line 175-177)
It would be important to add an image or figure that represents the procedure used. This would give a plus to the study because it would allow a better understanding of how the photographs were captured and then analyzed.
DISCUSSION (Line 276-288)
In this section of the discussion it would be important to be able to mention the values of HF angles and NF to establish the existing differences with other studies.
A subsection is suggested to specify in detail and perhaps point by point the limitations of the present study (Line 341-358)
These limitations such as the non-generalization of the results, the two-dimensional evaluation could serve as a reference for future studies.
It is therefore suggested to make this subsection more visible after the discussion.
A suggestion could also be to add as a limitation not to relate other variables or instruments that allow the identification of patterns or behaviors of the physical condition of the participants.
Finally, to review whether height could be a confounding variable in the study.
CONCLUSIONS (Line 360-371)
In the conclusions it would be necessary to detail the findings more precisely, defining that in response to gender there are differences for three variables (Neck flexion, Gaze angle and Vision), in hand operation only for head flexion and for Gender x Hand operation no differences were found.
With these inputs, the aim is to generate a better understanding of the head and neck movements performed by the participants in a real environment.
On the other hand, it would be essential to add two new sections or one that combines future research perspectives and practical applications of these findings. At the end it is detailed that the results cannot be generalized so more studies are needed, the question would be, what kind of studies, under what conditions, and which analyze what other possible variables. If they define this, it would also be a valuable contribution that would allow the present study.
Finally, I thank the authors for the excellent work and encourage you to review the comments shared.

Author Response
Thank you for your valuable comments and suggestions. Our detailed responses are provided in the attached document.

Reviewer 2 Report
Comments and Suggestions for Authors
The article makes an analyse of the gender-based differences in head and neck posture regarding the smartphone users when they walk. The study employ a single-blind observation method in natural environments to investigate the metrics such as neck flexion (NF), head flexion (HF), gaze angle (GA), and viewing distance (VD) with 120 participants (separated in 60 males and 60 females). The results that the authors obtain highlight significant differences in the metrics NF, GA, and VD between genders; this could suggest potential risks of musculoskeletal strain and eyestrain from habitual smartphone use.
Some key aspects of the article:
- Gender Differences – the males have shown a larger NF, GA, and VD compared to females.
- Risks Identified – the Forward head posture (FHP) in males could lead to increased neck and shoulder strain, while the shorter VD in females could exacerbate eyestrain.
- Hand Operations - Two-handed texting resulted in a larger VD than one-handed browsing.
- Real-World Insights - The study also emphasises the value of field observations in understanding natural postures, which is consistent with previous laboratory-based research.
Some Positive and Negative Aspects
Positive Aspects
- Single-blind field observations provide real-world relevance, which is a very innovative methodology.
- The fact that we have equal gender distribution enhances the reliability of gender comparisons.
- We have a detailed analysis, thorough statistical approach and use of validated measurement techniques.
Negative Aspects
- Since it excludes the three-dimensional posture assessments and the dynamic gait analysis, this may cause a limited scope.
- There are a lot of potential biases because uncontrolled variables (mentioned at the end of this review) could influence results.
- Observations in Taipei may not generalized to broader populations.
Ideas for the Future
- Incorporate three-dimensional motion analysis to capture complex postural dynamics.
- Expand the number of participants in order to include diverse age groups and geographical locations.
- Investigate the long-term health implications regarding the observation of the postural habits of the participants.
- Analyze other additional factors like the impact of the phone size and the participant distractions.
Author Response

(The authors gave the same response as above.)

Reviewer 3 Report
Comments and Suggestions for Authors
Please check the comments

Author Response

(The authors gave the same response as above.)

Round 2
Reviewer 3 Report
Comments and Suggestions for Authors Congratulations on the improvements made to the text.